# ARE LARGE LANGUAGE MODELS TRULY DEMOCRATIZING FINANCIAL KNOWLEDGE? IDENTIFYING KNOWLEDGE GAPS

## ABSTRACT

Large Language Models (LLMs) are frequently utilized as sources of knowledge for question-answering. While it is known that LLMs may lack access to real-time data or newer data produced after the model's cutoff date, it is less clear how their knowledge spans across *historical* information. In this study, we assess the breadth of LLMs' knowledge using financial data of U.S. publicly traded companies by evaluating more than 190k questions and comparing model responses to factual data. We further explore the impact of company characteristics, such as size, retail investment, institutional attention, and readability of financial filings, on the accuracy of knowledge represented in LLMs. Our results reveal that LLMs are less informed about past financial performance, but they display a stronger awareness of larger companies and more recent information. Interestingly, at the same time, our analysis also reveals that LLMs are more likely to hallucinate for larger companies, especially for data from more recent years. We will make the code, prompts, and model outputs public upon the publication of the work.

## 1 INTRODUCTION

As research and development of Large Language Models (LLMs) continues to progress, evidence has emerged of heavy usage of these models within the financial domain. Cheng et al. (2024) finds that trading volumes significantly decline during ChatGPT outages, suggesting heavy use of LLMs by investors, and numerous research papers have been published indicating that LLMs can provide investment advice or generate investment strategies (Romanko et al., 2023; Dong et al., 2024). Notably, Oehler & Horn (2024) finds that by offering advice on bond and equity ETFs, *"ChatGPT provides better financial advice for one-time investments than robo-advisors"*. More so, Yue et al. (2023) claims that LLMs can serve as a method to *democratize* financial knowledge. Such heavy reliance on LLMs in the financial industry warrants a closer investigation into the various biases that may be present in these models, as these biases may result in significant adverse affects when these models are employed for financial decisions. Biased or inaccurate information generated by LLMs may cause sub-optimal financial decision-making and lead to unintended downstream effects in financial markets. For example, as investors incorporate LLMs into their research workflows, a bias in LLMs towards favoring larger firms may ultimately cause reduced capital allocation to smaller firms (see Appendix K).

In the broad realm of natural language processing (NLP), there has been a considerable inquiry into biases, such as gender (Lu et al., 2020), political (Motoki et al., 2023), and cultural biases (Naous et al., 2024). However, there has been limited study of LLM biases in the financial domain. In this work, we aim to identify and analyze these biases by examining where knowledge gaps in these models are present. We consider how various factors such as time period, size, popularity among retail and institutional investors, market capitalization,[1] and the readability of financial reports affect the answering capabilities of LLMs, and whether these knowledge gaps are biased along certain temporal or cross-sectional[2] factors. Our study evaluates leading language models, including

---

[1] Definition and further reading for financial terminologies are provided in Appendix A.
[2] Cross-sectional analysis examines data from multiple companies at a single point in time.

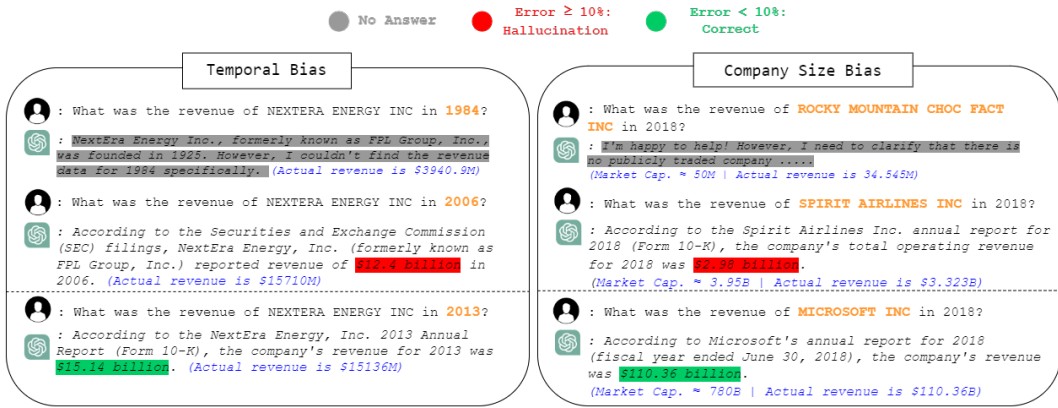

Figure 1: Example responses from Llama3-70B showcasing knowledge biases towards smaller (vs. bigger) companies and older (vs. newer) financial information.

ChatGPT (Achiam et al., 2023) and Llama-3-Chat (Touvron et al., 2023), revealing temporal and cross-sectional knowledge gaps in LLMs.

Our analysis shows several key findings. We see that there is a possible *"retrograde"* knowledge bias, where LLMs are unable to answer financial questions *before* a certain time (e.g., what is a company's revenue back in the year 1982? See Figure 1). This complements existing research that highlights LLMs' struggles with knowledge produced *after* the cut-off dates of pre-training data (Onoe et al., 2022; Kasai et al., 2024). For instance, Llama-3-70B-Chat accurately answers revenue values for 52.01% of companies in 2018 while answering accurately for only 7.47% of companies in the year 1995 —despite financial data being publicly available for all U.S. companies since 1995.

We also find that LLMs demonstrate better accuracy for companies with larger market capitalizations, higher attention from both retail and institutional investors, higher number of SEC filing accesses, and better filing readability. For Llama-3-70B-Chat, a tenfold increase in market capitalizations of the company leads to a 0.9551 rise in the log odds ratio of accurately answering revenue-related questions.

Additionally, we extend our analysis to examine how both time and firm factors relate to the model's propensity to produce *factuality hallucinations*, one of two types of hallucinations defined by Huang et al. (2023). Factuality hallucinations are defined as discrepancies between LLM-generated content and verifiable real-world facts. When the LLM response contains a numerical value, we can establish a numerical measurement to identify factuality hallucination. We calculate the error rates of LLM answers with respect to ground truth values and consider a response a hallucination if the error rate is above a certain threshold (10%). Interestingly, we find that LLMs tend to hallucinate more for those same firms for which it also sees higher accuracy; for Llama-3-70B-Chat, a tenfold increase in market capitalizations results in a 0.1712 rise in the log odds ratio of hallucinating revenue—again, despite financial data being widely available.

As a result of this work, we put forth the following contributions:

- Utilizing diverse financial datasets to construct and analyze over 190k questions, providing insights into the performance and bias of leading LLMs.

- Introducing a novel, systematic method for analyzing temporal knowledge biases of LLMs. To the best of our knowledge, this is the first study to analyze the retrograde knowledge bias in LLMs.

- Utilizing our cross-sectional analysis framework to investigate a range of factors that are associated with knowledge bias in LLMs.

- Conducting a comprehensive hallucination analysis that identifies a notable trend of over-confidence in LLMs, particularly evident in responses concerning specific companies and time periods.

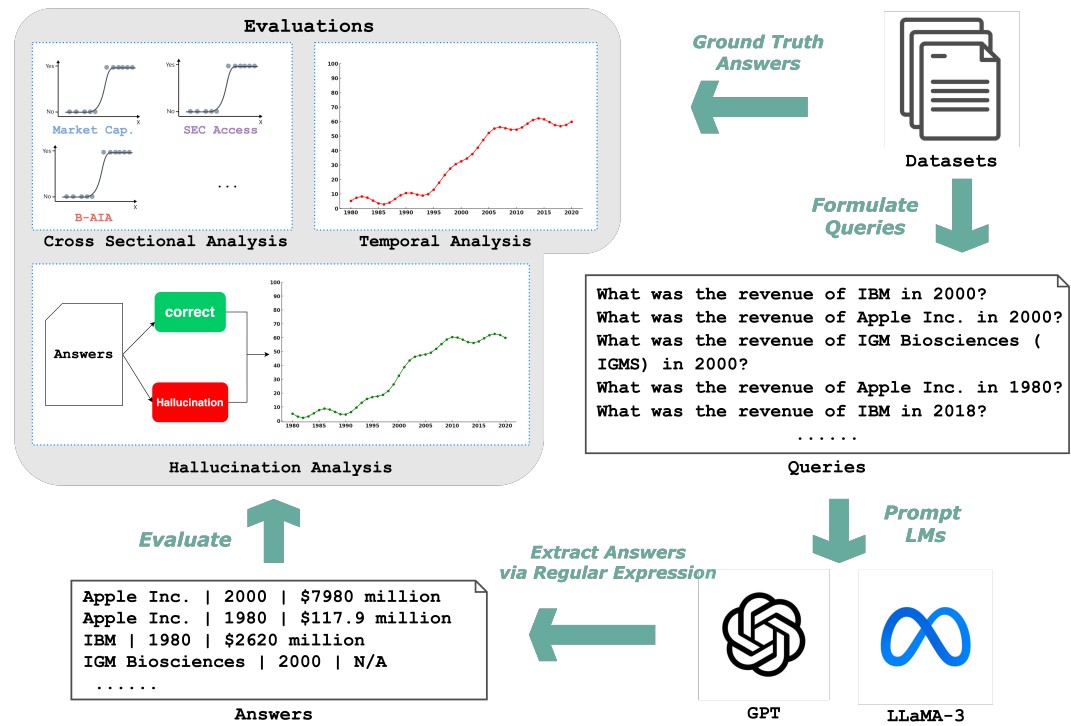

Figure 2: Experiment pipeline to understand and measure financial knowledge gap in LLMs.

## 2 EXPERIMENTAL SETUP

To facilitate our analysis, we begin by constructing RevenuePromptDataset: a novel dataset of over 190k question-answer pairs based on revenue data from more than 17k unique companies and spanning over 41 years (from 1980 to 2020).

### 2.1 CONSTRUCTION OF REVENUEPROMPTDATASET (RPD)

To construct our dataset, we compile 190,956 samples covering company-year samples available in both "Compustat North America" and "Monthly Stock File" databases from 1980 to 2020. Each sample has a company name, year, and revenue (in million USD) information. The dataset includes 17,621 unique companies, including those that have entered or exited the market due to IPOs, bankruptcy, or privatization.[3] Besides revenue and market cap, we also draw data from other resources to consider other key variables, including the number of retail investors, institutional attention, number of SEC filing access, and readability of regulatory filings by the company. These variables are utilized for our cross-sectional analysis, and as such we discuss these variables and their sources in more detail in Section 4.1. We provide supplementary details on each dataset in the Appendix B.1.

We use the entire dataset of 190k samples in most of our analyses for relatively less expensive models (GPT-3.5-turbo-0613, Llama-3-8B-Chat, and Llama-3-70B-Chat). For the costly GPT-4, and Gemini 1.5 Pro model, we use a subset ($RPD_{200perYear}$) of the full sample. In addition, we create another subset ($RPD_{460}$) with 460 companies that have a long public history, being available for every year from 1980 to 2020. These subsets of RPD are discussed in further detail in Appendix B.3.

### 2.2 QA PROMPT AND TASK DESIGN

We construct LLM prompts, as shown in Figure 2, programmatically by utilizing the following template: *What was the revenue of {company_name} in {financial_year}?*

---

[3] Figure 6 in Appendix B.2 shows the change in the number of companies over time in our dataset.

This study's prompt design is informed by the observation that retail investors with limited financial sophistication are inclined to rely on readily accessible LLMs. Given the potential for errors or inaccuracies in these models, such vulnerable populations could face significant adverse consequences. This concern is underscored by the analysis presented in Chava et al. (2022), which examines the responses of financially unsophisticated investors to non-informative cryptocurrency endorsements by celebrities. While more elaborate and sophisticated prompting strategies may yield more precise outcomes, it is less likely that these investor groups would employ complex prompting methods or engage in further fine-tuning of the models. Moreover, the LLMs are able to understand the simple prompting employed here. This is verified in the manual evaluation of the small sample presented in Appendix D.

We utilize our prompts to perform zero-shot prompting on the three LLMs. For our three LLMs, we utilize a $temperature$ value of 0.00 (for reproducibility) and $max\_token$ value of 100. Further implementation details for models are provided in Appendix C. Based on the output of the model, we utilize regex to extract the numerical value of revenue if it is available in the answer, and to standardize units to millions of dollars for consistency in evaluations we perform (Figure 3). The effectiveness of regex is discussed in Appendix D. We calculate absolute % error between the model's extracted revenue value and the ground truth revenue value we have. We utilize this calculation for temporal and cross-sectional analysis later.

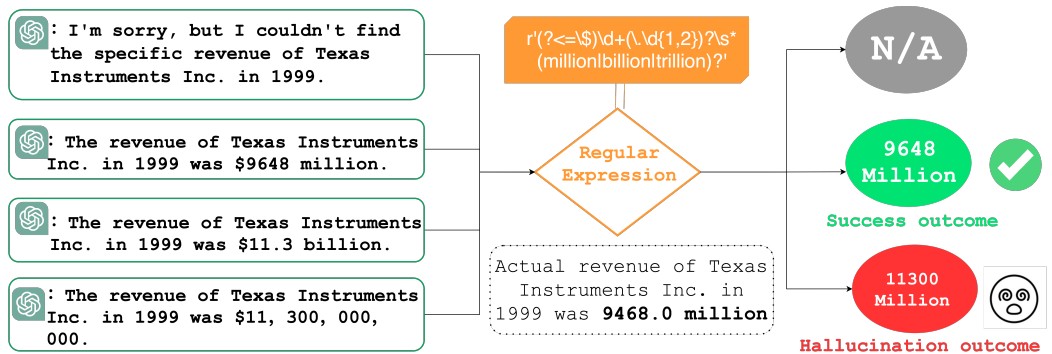

Figure 3: Extract revenue information from LLMs answer via regular expression.

## 3 TEMPORAL FINANCIAL KNOWLEDGE GAPS IN LLMS

Since 1995, annual regulatory filings of publicly traded companies in the U.S. have been transparently available in the Securities and Exchange Commission (SEC)'s EDGAR[4] database. Given this, we postulate that the models might have an ascending knowledge gradient from 1995 onwards. Figure 1 (left side) using NextEra Energy, Inc. as an illustrative example supports this hypothesis. Utilizing our prompting dataset (RPD), we begin our analysis by examining temporal knowledge gap bias of three LLMs (Llama-3-Chat-8B, Llama-3-Chat-70B, and GPT-3.5) for their answering accuracy on our 190k revenue questions.

### 3.1 METRICS FOR TEMPORAL ANALYSIS

For temporal analysis, we create a ternary outcome variable $Y_{i,t}$ for firm $i$ and year $t$ based on the model's answer to evaluate the model performance. The variable $Y_{i,t}$ takes the value 2 if an absolute % error of the answer is less than 10% (Refer Appendix E), 1 if an absolute % error of the answer is greater than 10% and 0 if no numeric answer is provided. We consider a value of 2 as success which is used to calculate the model's success rate for all companies over the years. On the other hand, we consider a value of 1 for $Y_{i,t}$ as factual hallucination by the model, and we use it for calculating the rate of hallucination.

---

[4] https://www.sec.gov/edgar/search-and-access

$$Y_{i,t} = \begin{cases} 2: & \text{absolute \% error} < 10\% \\ 1: & \text{absolute \% error} \geq 10\% \\ 0: & \text{no numerical answer} \end{cases} \quad (1)$$

We employ Equation 2 below to calculate the success rate and hallucination rate for a given year (T) and analyze the temporal trend over the years.

$$\text{Success Rate (T)} = \frac{\sum_{i,t=T} \mathbb{1}_{\{Y_{i,t}=2\}}}{\sum_{i,t=T} 1}$$

$$\text{Hallucination Rate (T)} = \frac{\sum_{i,t=T} \mathbb{1}_{\{Y_{i,t}=1\}}}{\sum_{i,t=T} 1} \quad (2)$$

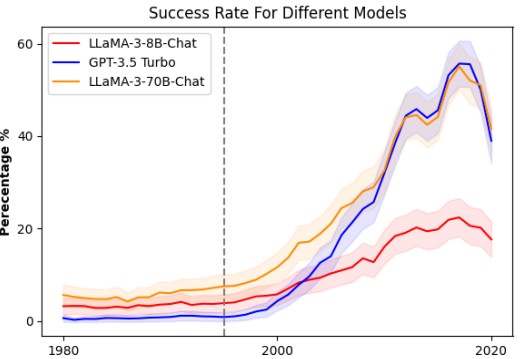 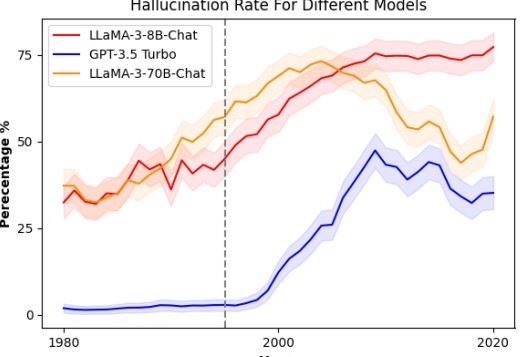

Figure 4: Success and Hallucination rate of GPT-3.5 Turbo ("gpt-3.5-turbo-0613") and Llamas ("Llama-3-8B-Chat", and "Llama-3-70B-Chat") over time. The dotted line is drawn at the year 1995. The shadow area around the line is the standard deviation of model performance.

### 3.2 RESULTS AND ANALYSES

In Figure 4 (left), we present the temporal success rate trends of three models. This data underscores our first claim: LLMs demonstrate a heightened proficiency in answering question from more recent years as opposed to earlier ones. It's pertinent to highlight the dotted line at the year 1995, signifying the inception of the SEC's EDGAR filing system. After this date, detailed financial information from US public companies became publicly accessible online, thus augmenting the datasets available for model training. Intriguingly, there is a noticeable dip in the performance of all models in the years 2019 and 2020 when compared against their performance in 2018. Similar results for GPT-4, and Gemini 1.5 Pro on RPD$_{200perYear}$ sample are presented in Figure 8 of Appendix F. Investigating the underlying reason behind this deviation promises to be a compelling avenue for further research.

In Figure 4 (right), we present the percentage of companies (hallucination rate) for which the model hallucinates the revenue information over the years. Among the three models, Llama-3-8B-Chat hallucinates at a much higher rate over recent years. The hallucination trends for Llama-3-70B-Chat and GPT-3.5 Turbo models share similarities to their success trends. Such similarity demonstrates that Llama-3-70B-Chat and GPT-3.5 Turbo models are more likely to hallucinate for the same years for which they are also more likely to provide the correct answer.

## 4 MEASURING LLMS' CROSS-SECTIONAL BIASES IN FINANCE

In addition to temporal knowledge gap biases, we also explored how various cross-sectional factors affect the distribution of knowledge gaps. In financial literature, there have been studies showing a relation between analysts' biases and the size of the firm (Van Binsbergen et al., 2023) and role of institutional attention (Ben-Rephael et al., 2017) in helping incorporate information in asset prices.

Nevertheless, to the best of our knowledge, there is no study analyzing the relationship between firm-level information (size, institutional attention, popularity, etc.) and knowledge biases in LLMs. We extend our analysis to address this research gap and further explore how cross-sectional factors affect LLM knowledge and hallucination rates.

| Data (Source) | Key Variables | Years | Sample Size | Public |
|---|---|---|---|---|
| Annual Financials (Compustat Capital-IQ) | Revenue | 1980-2021 | 422,792 | ✗ |
| Market Capitalization (CRSP MSF) | Price and # Shares | 1980-2021 | 333,220 | ✗ |
| Robinhood (Robintrack) | # Holders | 2018-2020 | 23,337 | ✓ |
| Bloomberg AIA (Bloomberg) | B-AIA | 2010-2020 | 25,224 | ✗ |
| SEC Access (SEC-EDGAR) | # Access | 2003-2017 | 10,631,960 | ✓ |
| Bog Index (Bonsall IV et al., 2017) | Bog Index | 1994-2021 | 189,076 | ✓ |

Table 1: We utilize a wide range of finance datasets from notable sources.

## 4.1 CROSS-SECTIONAL VARIABLES

We utilize various datasets to evaluate the influence of factors like market capitalization and other company-level variables (controlling for year) on the knowledge bias of LLMs. These include market capitalization, retail investment using Robintrack[5] data, institutional attention as indicated by Bloomberg's Abnormal Institutional Attention (AIA), the readability of SEC filings[6] (e.g., active voice, fewer hidden verbs, etc.) as measured by the Bog Index, and the frequency of regulatory filing downloads through SEC access log files. By examining these diverse and multifaceted factors, we aim to shed light on the various elements that potentially shape the knowledge bias in LLMs. The dataset are summarized in Table 1.

**Market Capitalization (MCap):** We collect market-related information such as price, number of outstanding shares, and identifiers from the "Monthly Stock File" database of The Center for Research in Security Prices, LLC (CRSP). We have access to datasets through WRDS. We cover the data starting in 1980 and ending in 2021. After collecting the data we calculate the market capitalization for a company for a given day by multiplying the number of outstanding shares with the price of the share on the day. For a particular financial year, we use the market cap value on the last trading day as a value for that year. Inflation adjustments and scaling for the market cap are discussed in Appendix B.1.

**Robinhood Retail Attention:** To understand how the popularity of the stocks among retail investors correlates with the model's ability to answer the question for those companies, we collect the data for Robinhood from Robintrack[7]. In particular, we collect the popularity metric which represents the number of unique accounts that hold at least one share of the stock. To ensure comparability, we apply standardization to the data, involving mean subtraction followed by division by the standard deviation. The data has Ticker as an identifier, which can be used for matching with other datasets. We have this data available only from 2018 to 2020.

**Bloomberg Abnormal Institutional Attention (B-AIA):** Our measure of institutional attention, sourced from Bloomberg, caters to a limited audience of around 320,000 subscribers, mainly institutional investors, due to the high cost of $24,000 per annual subscription. We collect the data from Bloomberg Terminal which we have accessed through our institution. In our analysis, we utilize methodology used in Chava & Paradkar (2016) to measure attention on specific stocks, based on views of news articles and search activities. By aggregating the highest hourly scores into a daily attention score, we gain insights into the institutional investor interest. We apply standardization to the data similar to Robinhood data. Refer to Appendix B.1 for more details.

---

[5]Robintrack keeps track of how many Robinhood users hold a particular stock over time.

[6]SEC filings include financial statements and other formal documents submitted to the U.S Securities and Exchange Commission (SEC).

[7]https://robintrack.net/

**SEC Access:** As a proxy for a company's popularity among investors, we utilize over 15 billion searches on SEC.gov from February 2003 to June 2017, recorded in the EDGAR Log File,[8] converting queries into CIK–year pairs to measure average daily filing accesses. We apply standardization to the data similar to Robinhood data.

**Bog Index for Readability:** We also consider the readability of the financial reports companies filed with the SEC between 1994 and 2021 to explore how variations in readability may correlate with language models' ability to answer on those companies. We use a commonly used dataset[9] that contains 189,076 10-K fillings from 20,936 companies and their readability scores calculated by the Bog Index Bonsall IV et al. (2017). The higher score equates to a less readable document.

## 4.2 METRICS FOR CROSS-SECTIONAL ANALYSIS

We use the same ternary outcome variable $Y_{i,t}$ that indicates model performance as discussed in Section 3.1 for our temporal analysis. To capture the relationship between the model's answer and company-level characteristics, we run the following logistical regression with $X_{i,t}$ as company-level characteristics:

$$logit(P(Y_{i,t} = y)) = \alpha + \beta * X_{i,t} + \delta_t * D_t + \epsilon_{i,t} \qquad (3)$$

Here $Y_{i,t}$ is the outcome variable where $y = 2$ indicating model success while $y = 1$ indicating hallucination, $\delta_t$ is a year-fixed effect, $\alpha$ is a constant term, and $\epsilon_{i,t}$ is an error term. The coefficient $\beta$ will help us understand the influence of company-level characteristic $X_{i,t}$ on the outcome variable ($Y_{i,t}$). We use the above regression to analyze the outcome of all models.

## 4.3 RESULTS AND ANALYSES

| $X_{i,t}$ | Constant ($\alpha$) | Beta ($\beta$) | Constant ($\alpha$) | Beta ($\beta$) | Constant ($\alpha$) | Beta ($\beta$) |
| --- | --- | --- | --- | --- | --- | --- |
| | Llama-3-8B | | Llama-3-70B | | GPT-3.5 Turbo | |
| MCap (log) | -11.5375‡ | 0.9521‡ | -10.9497‡ | 0.9551‡ | -16.0714‡ | 1.2647‡ |
| retail_inv (std) | -1.2511‡ | 0.1940‡ | 0.2166‡ | 0.1080‡ | 0.3966‡ | 0.3341‡ |
| B-AIA (std) | -1.0392‡ | 0.0262 | -0.1842‡ | 0.0116† | -0.1185† | 0.0333† |
| SEC-Access (std) | -2.3396‡ | 0.0671‡ | -1.5708‡ | 0.0782‡ | -2.2310‡ | 0.1163‡ |
| Bog Index (std) | -2.3901‡ | -0.0904‡ | -1.7725‡ | -0.0746‡ | -3.1495‡ | -0.1174‡ |

Table 2: Empirical cross-sectional regression results for market cap (MCap), number of retail investors on Robinhood (retail_inv), Bloomberg abnormal institutional attention (B-AIA), number of access on SEC-EDGAR (SEC-Access), and measure of readability (Bog Index). "std" in the parentheses indicates the data is standard normalized. *, †, and ‡ indicate significance at the 10%, 5%, and 1% levels, respectively. The results are based on the full sample with year fixed effect.

To investigate the influence of a company's market capitalization (size) on the proficiency of LLMs in answering the company's financial details, we conducted a logistic regression analysis as outlined in Equation 3. In this regression, we adopted the logarithm (base 10) of the company's market cap as the independent variable, denoted as $X_{i,t}$. The outcomes of this regression analysis are presented in Table 2. The data suggests that LLMs exhibit enhanced performance for companies with larger market capitalization. A tenfold increase in the market cap corresponds to an increment of 1.2647 in the log odds ratio

| Model | Constant ($\alpha$) | Beta ($\beta$) |
| --- | --- | --- |
| GPT-3.5 Turbo | -6.3148‡ | 0.2825‡ |
| Llama-3-8B | -4.6140‡ | 0.4696‡ |
| Llama-3-70B | -1.9279‡ | 0.1712‡ |

Table 3: Market cap analysis results on hallucination based on the empirical regression. *, †, and ‡ indicate significance at the 10%, 5%, and 1% levels, respectively. The results are based on the full sample with year fixed effect.

---

[8]See https://www.sec.gov/about/data/edgar-log-file-data-sets for more details.
[9]https://host.kelley.iu.edu/bpm/activities/bogindex.html

of the GPT-3.5 Turbo model answering revenue-related questions correctly, and an increase of 0.9521 and 0.9551 for the Llama-3-8B-Chat and Llama-3-70B-Chat models respectively. Similar results for GPT-4 and Gemini 1.5 Pro on RPD$_{200perYear}$ sample are presented in Table 6 of Appendix F. The Results of Bloomberg AIA, SEC Access, and retail investment from Robinhood data follow a similar trend. As a higher value of the Bog Index represents that the filing is less readable, the variable has a negative beta indicating that the lower readability leads to a decrease in the performance of LLMs.

We similarly regress hallucination on the log market cap using Equation 3 by setting y=1. Based on the results in Table 3, we find that the model is more likely to hallucinate for the companies with higher market cap. Similar results for GPT-4 and Gemini 1.5 Pro on RPD$_{200perYear}$ sample are presented in Table 6 of Appendix F. Combining the result of Table 2 and Table 3, we can hypothesize that the model is more likely to hallucinate for the same companies for which it is also more likely to provide the correct answer (have knowledge).

For further analysis, we count the number of years for which a given company produces LLM hallucinations and the number of years for which it provides the correct answer. The visualization of such a relationship for GPT-3.5 Turbo in Figure 5, shows that the revenue of companies with a higher market cap (indicated with a color close to yellow) have a higher propensity to be answered correctly and hallucinated at the same time. Similar results for Llama-3-70B-Chat are provided in Figure 11 of Appendix H for the robustness of the finding. We hypothesize these results might be due to the model having higher confidence for relatively recent years and those higher market cap companies.

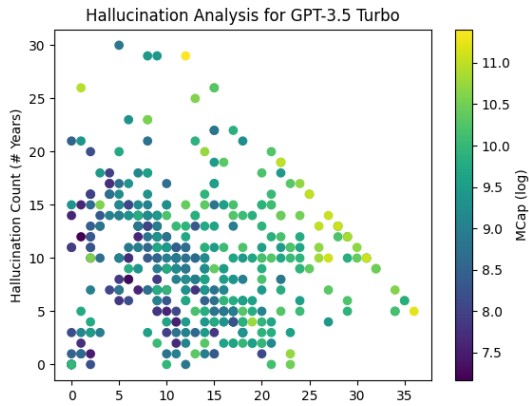

Figure 5: Each dot represents a company with a count on the x-axis indicating the number of years for which ChatGPT-3.5 Turbo gave the correct answer and the y-axis indicating the number of years for which ChatGPT-3.5 Turbo hallucinated in the answer. We take an average of market cap to assign a color of the dot.

## 5 DISCUSSION

Our results highlight significant biases in the financial knowledge gaps of LLMs, along both temporal and cross-sectional dimensions. These biases are likely to have a substantial effect on the efficacy of LLM use both in the financial domain and across other domains as well. In addition, as LLMs continue to be touted as a *"democratizing"* force for financial knowledge, it is important to consider what effects biases may have on this process.

**Investors vs. Researchers:** It is important for investors who continue to utilize LLMs for their workflows to be aware of the risks for bias we highlight in our work. We caution investors to not rely on LLMs for factual information and/or open-ended analysis, as the biases demonstrated in our work may cause unintended harm. In particular, the effects of these biases may cause more harm for new investors, whose inexperience may lead them to be less likely to discern incorrect factual information produced by LLMs. For example, Kumar (2009) shows that younger investors tend to invest more in "lottery-type" stocks, sacrificing financial due diligence and leaving them more vulnerable to biases in LLM-generated investment advice. We encourage financial literacy educators to consider the harmful effects of LLM bias, and make these risks clear in their teachings.

Researchers who study applications of LLMs in finance should be mindful to take into consideration the various stratas that their particular task may span, and design their study and experiments to include comprehensive coverage of those stratas. They should ensure their experimentation covers firms (or other entities) across a breadth of years, sizes, degrees of prominence, and geographic or cultural boundaries. Careful consideration of stratified biases may add to the strength of results and

ensure that results were not the result of biases in the model, or that the results were only valid for certain stratas.

**Guidance for LLM developers:** LLM developers may be able to, and should to the best of their ability, take measures that may reduce bias in their models. Most importantly, they should be mindful of the data used for pre-training and consider how biases in this data may cause biases in the model. Such analysis of pre-training data sources is especially relevant for domain-specific models. While it may be more difficult to assess all potential sources of bias for LLMs trained on a general-domain corpus, developers of domain-specific LLMs should keep in mind the narrower set of potential use cases for their model and ensure that pre-training data is optimized to reduce bias in those use cases. In addition, debiasing techniques can also be employed to further improve the efficacy of domain-specific models.

**Broader Impacts and Future Work:** While we focus on the financial domain in this work, our temporal and cross-sectional evaluation and analysis framework is broadly applicable for assessing LLM knowledge across various domains. Many knowledge-intensive tasks for which LLMs may be utilized can be assessed both temporally and along certain entity-specific stratas. For example, an evaluation of LLM applications in the legal domain may examine LLMs' ability to answer questions about laws passed at different times (e.g. can the LLM answer questions better about a law passed in 2018 compared to a law passed in 1890?), as well as assessing LLM ability to answer questions about laws with varying media attention.

The application of our temporal and cross-sectional analysis frameworks to other domains in which LLMs are frequently utilized would be of significant value. Knowledge biases are likely to cause significant adverse affects in most fields, and successfully identifying such biases in other fields is a promising avenue for future research.

As an example of an extension of our framework for more diverse applications, we apply it to assess LLMs' ability to answer questions about soccer statistics from the annual La Liga championship. We utilized our evaluation framework to assess Llama3-70B's ability to answer the following question: *"In the {years} La Liga season, how many points did {team} finish with?"*. We find that the model sees higher accuracy for more recent seasons and for more prominent soccer clubs, highlighting a potential bias towards more recent seasons and larger clubs. We hope that such analysis provides a clear demonstration of the value that our evaluation framework can bring to other domains. We present an in-depth description of our methodology and results for the soccer experiments in Appendix L.

## 6 RELATED WORK

We provide a brief review of the existing research regarding societal and political biases and hallucination in LLMs. Our study is distinctive in its approach, as it is, to our understanding, the inaugural effort to retrospectively assess knowledge bias in LLMs. Furthermore, this research pioneers in exploring the multitude of factors influencing financial knowledge bias within these models.

**Biases in LLMs:** There has been much research involving the evaluation of LLMs' societal bias, such as gender, religion, race, politics, etc (Zhao et al., 2017; Lu et al., 2020; Sheng et al., 2021; Nozza et al., 2022; Kotek et al., 2023; Zhao et al., 2023; Blodgett et al., 2020). Inspired by software testing, Nozza et al. (2022) proposed a systematic way of integrating societal bias testing into development pipelines. Zhao et al. (2023) unraveled the LLMs' "re-judgment inconsistency" in bias evaluation by leveraging psychological theories. Gender bias, as the issue most frequently addressed, has been resolved by diversified methods. Zhao et al. (2018) curated a new benchmark dataset WinoBias for testing and dispelling gender bias in LLMs. Kotek et al. (2023) use a simpler paradigm to identify gender bias and reveal that LLMs still tend to reflect the imbalance of their training dataset even after aligning with human preference. Additionally, Motoki et al. (2023) extends bias evaluations to political domains, revealing significant biases in ChatGPT towards certain political groups, necessitating further scrutiny measures in training processes.

**Hallucination in LLMs:** Research on measuring hallucination in LLMs has produced several innovative methodologies aimed at quantifying and addressing the issue. Zellers et al. (2019) were among the pioneers, introducing Grover, a model designed for both generating and detecting "neural

fake news" to study the propensity of LLMs to generate hallucinated content. Goyal & Durrett (2020) introduces a method to assess the factuality of text generation, targeting hallucinations by analyzing the dependency-level entailment between generated content and source material. Lin et al. (2021) provides a dataset for testing model truthfulness, indirectly evaluating hallucinations by measuring alignment with factual information. Huang et al. (2023) presents a comprehensive overview of hallucination detection methods and benchmarks, categorizing hallucination into two groups: factuality hallucination and faithfulness hallucination. To the best of our knowledge, our work is the first to measure hallucination in LLMs for the financial domain.

**LLMs in Finance:** Recent advancements in LLMs have significantly impacted the financial domain. Nie et al. (2024) introduced CFinBench, a comprehensive Chinese financial benchmark designed to assess LLMs' financial knowledge across various categories, including financial subjects, qualifications, practices, and laws. Similarly, Xie et al. (2023) developed PIXIU, a framework comprising a financial LLM fine-tuned with instruction data, alongside an evaluation benchmark covering various financial tasks. Kosireddy et al. (2024) explored the readiness of small language models for democratizing financial literacy, analyzing their performance in financial question-answering tasks. Additionally, Shah & Chava (2023) benchmarked the zero-shot performance of LLMs like ChatGPT on financial tasks, comparing them with fine-tuned models to assess their effectiveness in the financial domain. Collectively, these studies underscore the growing role of LLMs in enhancing financial applications and literacy.

## 7  CONCLUSION

In this study, we examine both the temporal knowledge bias and bias across firm-characteristic stratas in LLMs using financial data from U.S. publicly traded companies. Our findings reveal that LLMs are more proficient with recent financial information, especially after the 1995 introduction of the SEC's EDGAR filing system. However, there's an unexplained dip in performance for 2019 and 2020. Secondly, LLMs demonstrate better accuracy for companies with larger market capitalizations, higher retail investment, higher institutional attention, higher number of SEC filing access, and higher readability. We also find that in the years and companies for which LLMs are more likely to provide a correct answer, they are also more likely to hallucinate. In essence, while LLMs offer valuable insights, their limitations in financial knowledge necessitate careful usage, especially in professional financial domains. Future work should explore the reasons behind these trends and enhance LLMs' performance breadth, as well as expand our methodology to other domains. This study further contributes to the discourse on how the availability of the pre-training corpus of LLMs for independent scientific scrutiny can facilitate scientific advancement.

## LIMITATIONS

We do not run the analysis for GPT-4 and Gemini 1.5 Pro on the full sample due to high API cost. While we provide a small analysis for Llama-based financial LLMs in Appendix G, We are unable to run analysis on finance domain-specific pre-trained LLMs like BloombergGPT (Wu et al., 2023) as these models are not publicly available. We acknowledge that "revenue" might not be relevant for every investor or researcher, but it provides a good proxy to understand temporal and cross-sectional biases in LLMs. More discussion on why we choose "revenue" as a question is provided in the Appendix J. We also discuss Ethical considerations of our work in Appendix M.

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

## A  Financial Lexicon: Definitions and Further Reading

Below we provide standard definitions of the finance-related lexicon used in the paper with the reference to further readings.

- **Outstanding Shares**: The total number of shares that are currently owned by all shareholders, including share blocks held by institutional investors and restricted shares owned by the company's officers and insiders. Further reading: https://www.investopedia.com/terms/o/outstandingshares.asp

- **Market Capitalization (MCap)**: The total market value of a company's outstanding shares of stock. It is calculated by multiplying the current market price of one share by the total number of outstanding shares. Further reading: https://www.investopedia.com/terms/m/marketcapitalization.asp

- **Consumer Price Index (CPI)**: A measure that examines the weighted average of prices of a basket of consumer goods and services, such as transportation, food, and medical care. It is calculated by taking price changes for each item in the predetermined basket of goods and averaging them. Further reading: https://www.investopedia.com/terms/c/consumerpriceindex.asp

- **Revenue**: The total amount of money generated by the sale of goods or services related to the company's primary operations. Further reading: https://www.investopedia.com/terms/r/revenue.asp

- **Robinhood**: A brokerage firm that offers commission-free trading of stocks and exchange-traded funds. Further reading: https://robinhood.com/us/en/about-us/

- **Ticker**: A unique series of letters assigned to a security for trading purposes. It is also known as a stock symbol. Further reading: https://www.investopedia.com/terms/s/stocksymbol.asp

- **IPO (Initial Public Offering)**: The process by which a private company can go public by the sale of its stocks to the general public. It could be a new, young company or an older company that decides to be listed on an exchange and hence goes public. Further reading: https://www.investopedia.com/terms/i/ipo.asp

- **Bankruptcy**: A legal proceeding involving a person or business that is unable to repay their outstanding debts. The process begins with a petition filed by the debtor or on behalf of creditors. Further reading: https://www.investopedia.com/terms/b/bankruptcy.asp

- **Privatization**: The transfer of a business, industry, or service from public to private ownership and control. Further reading: https://www.investopedia.com/terms/p/privatization.asp

## B  Supplementary Data Information

### B.1  Dataset Additional Details

- **Compustat Capital-IQ**: We only keep companies that provide consolidated (i.e. combined accounts for parent and subsidiary) financial statements and which report data in "Standardized" format according to Compustat – Capital IQ. This will exclude some companies located outside the U.S. (but listed in the U.S.) as they are not required to report consolidated financial statements.

- **Inflation Adjustments and Scaling for Market Capitalization**: For better scaling when running analysis, we convert the market capitalization values to log values with a base of 10. We then normalize these values to adjust for inflation using the Consumer Price Index (CPI) data collected from FRED (Federal Reserve Economic Data). The inflation-adjusted market cap is converted to values corresponding to December 2021. The formula used to adjust the market cap values of company $i$ on date $t$ for inflation is as follows:

$$MCap_{i,t,r} = MCap_{i,t,t} * \frac{CPI(r)}{CPI(t)}$$

  where $MCap_{i,t,r}$ represents the adjusted market capitalization of company $i$ on date $r$, $MCap_{i,t,t}$ denotes the market capitalization of company $i$ as measured on date $t$, while $CPI(r)$ and $CPI(t)$ are the CPI for dates $r$ and $t$ respectively.

- **Robintrack**: In this data when we share it only includes normal, long shares (not options).

- **B-AIA**: The methodology employed to quantify institutional investor attention involves a process, as outlined by Bloomberg. Each news article read is assigned a score of 1, while searches are weighted more heavily at a score of 10. These activities are aggregated on an hourly basis, and the attention score is derived by comparing these hourly counts to the previous month's average, with adjustments for deviation levels. Scores range from 0 to 4, reflecting varying degrees of attention based on the percentiles of activity compared to the prior month. This process effectively captures the intensity of investor focus on a stock, with daily scores determined by the peak hourly attention score. For additional methodological details, refer to Chava & Paradkar (2016).

We use the following identifiers from datasets to merge them for analysis:

- Compustat Capital-IQ: GVKEY, Company Name, Ticker, CIK[10]
- CRSP MSF: PERMNO, Ticker, Company Name
- Robintrack: Ticker
- B-AIA: Ticker
- SEC Access: CIK
- Bog Index: GVKEY

### B.2  Companies Over Time

Figure 6 shows how the number of companies in our sample changed over time between 1980 and 2020. The year 1997 had the highest number of public companies during the dot-com bubble.

### B.3  Data Samples

The sample size for each sample is listed in Table 4.

---

[10]The Central Index Key (CIK) is a unique identifier used by the SEC to identify filing companies.

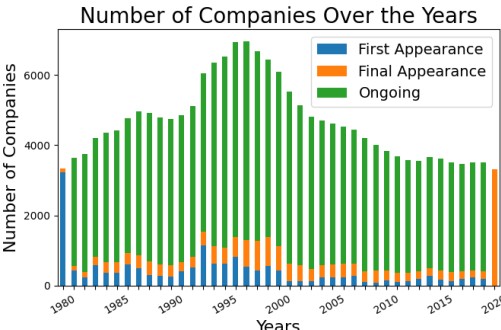

Figure 6: Number of public companies in the United States in our merged sample over time for 1980 to 2020 period. The first indicates the number of companies appearing in our sample for the first time, last indicates it appears for the last time. Companies that appear only for a single year are counted in the "Last" category and not in the "First" category.

**RPD$_{200perYear}$:** Given the high API cost for GPT-4 and Gemini 1.5 Pro, we create a subset representative sample of our data. To do so we categorize the log market cap of the companies into 4 categories (i.e., <8.00, 8.xx, 9.xx, >=10.00)[11]. After that, we randomly sample 50 companies from each year and market cap categories making it 200 samples per year. For 41 years, the total number of samples will be 8,200 (200*41). The result of GPT-4 and Gemini 1.5 Pro on RPD$_{200perYear}$ sample is presented in Appendix F.

**RPD$_{460}$:** As new companies can go public and existing public can get delisted, the set of companies changes every year. To analyze the same set of companies over time, we created a sample of companies that were public for every year from 1980 to 2020. We have 460 companies in our sample. For 41 years, the total number of samples will be 18,860 (460*41). The result of these three models on RPD$_{460}$ sample is presented in Appendix I for the robustness check.

| Sample Type | Sample Size |
|---|---|
| Full Sample (RPD) | 190,956 |
| RPD$_{200perYear}$ | 8,200 |
| RPD$_{460}$ | 18,860 |

Table 4: Sample size for different samples used in our analysis.

## C MODEL IMPLEMENTATION DETAILS

All the API calls are made between Oct 20, 2023, and Oct 26, 2023, for "gpt-3.5-turbo-0613" and between Nov 28, 2023, and Nov 29, 2023 for "gpt-4-0613". For "gemini-1.5-pro" we call API on Nov 18, 2024. We ran inference on "Llama-3-8B-Chat" locally using Transformer (Wolf et al., 2020) library on NVIDIA RTX A40 GPU. For the "Llama-3-70B-Chat" inference, we use API from together.ai. We are grateful to them for providing free credits and making it possible.

## D MANUAL VERIFICATION

We manually check the correctness of our prompting and regular expression in extracting revenue information from LLMs' outputs. We randomly sampled 100 ChatGPT's answers in 2010. There are 85 numerical answers (answers containing numerical revenue) and 15 no-answers (answers containing no revenue). Our regular expression returns *None* for all no-answers and correctly retrieves all numerical revenue. For all 100 samples, we observe that the LLM does not have difficulty understanding the question.

---

[11]Here 8.00 corresponds to $100 million in 2021 value.

# E    ROBUSTNESS ERROR THRESHOLD

We repeat both temporal and cross-sectional analysis by varying the error threshold of 5%, 10%, and 20%. The results in Figure 7 and Table 5 show that our findings are consistent.

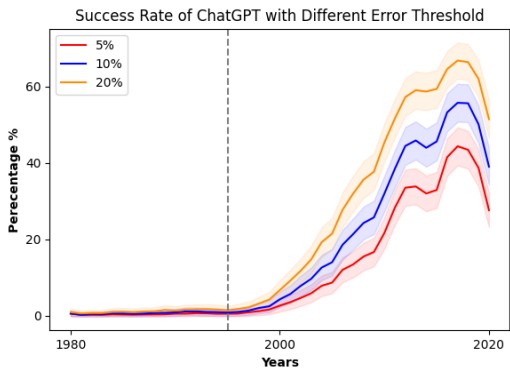

Figure 7: Performance of GPT-3.5 Turbo ("gpt-3.5-turbo-0613") for three different error thresholds over time. The dotted line is drawn at the year 1995. The shadow area around the line is the standard deviation of model performance.

| Error Threshold | Constant ($\alpha$) | Beta ($\beta$) |
|---|---|---|
| 5% | -16.1728‡ | 1.2577‡ |
| 10% | -16.0714‡ | 1.2647‡ |
| 20% | -16.1659‡ | 1.3242‡ |

Table 5: Market cap analysis results on correctness based on the empirical regression for GPT-3.5 Turbo for three different threshold values of error. *, †, and ‡ indicate significance at the 10%, 5%, and 1% levels, respectively.

We calculate the raw error of GPT-3.5 Turbo over time and plot it in Figure 14. The color bar indicates the percentage of companies for which GPT-3.5 outputs an answer containing numerical revenue information for each specific year. The black dots outside the box are the outliers, representing GPT-3.5's hallucination. There are many more outliers in the year where GPT-3.5's answer rate is high, which is consistent with our findings in hallucination analysis: GPT-3.5 model is more likely to hallucinate for the year that it is also more likely to provide the correct answer. The overall trend of raw error also aligns with the error trend in our temporal analysis.

# F    GPT-4 AND GEMINI 1.5 PRO ON SMALL SAMPLE

In Figure 8, we compare the performance of Gemini ("gemini-1.5-pro"), ChatGPT ("gpt-3.5-turbo-0613-0613"), GPT-4("gpt-4-0613"), Llama-3-8B ("Llama-3-8B-chat") and Llama-3-70B ("Llama-3-70B-chat") over time. The performance is measured for all four models on RPD$_{200perYear}$ sample. Surprisingly, the performance of ChatGPT is higher than that of GPT-4 for most years. GPT-4 has the best performance in 2018 outperforming GPT-3.5 by a small margin.

# G    FINMA RESULTS

In the Figure 9, we compare the temporal performance trend of FinMA-7B-full[12] with other models on RPD$_{200perYear}$ sample. The results show that FinMA performs the worst among all the models tested. It would be an interesting future study to explore the reasons behind the lower performance of the finance-domain instruction-tuned models.

---

[12]https://huggingface.co/ChanceFocus/finma-7b-full

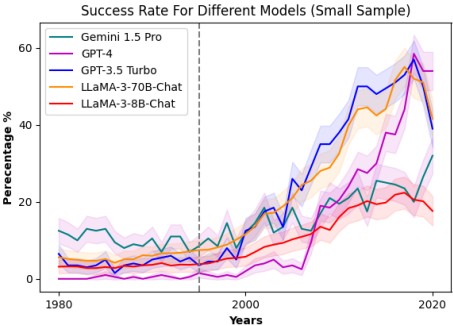

Figure 8: Performance of Gemini ("gemini-1.5-pro"), ChatGPT ("gpt-3.5-turbo-0613-0613"), GPT-4("gpt-4-0613"), Llama-8B ("Llama-3-8B-chat") and Llama ("Llama-3-70B-chat") over time. The dotted line is drawn at the year 1995. The performance is measured for all four models on RPD$_{200perYear}$ sample.

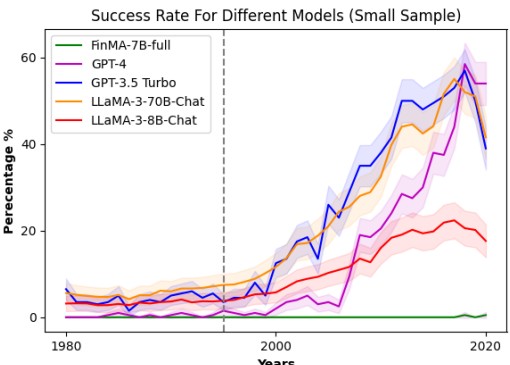

Figure 9: Success rate comparison of FinMA (FinMA-7B-full) with ChatGPT ("gpt-3.5-turbo-0613-0613"), GPT-4("gpt-4-0613"), Llama-3-8B ("Llama-3-8B-chat") and Llama-3-70B ("Llama-3-70B-chat") over time. The dotted line is drawn at the year 1995. The performance is measured for all four models on RPD$_{200perYear}$ sample.

## H HALLUCINATION ADDITIONAL RESULTS

| Analysis | Model | Constant ($\alpha$) | Beta ($\beta$) |
|---|---|---|---|
| Success | GPT-4 | -37.7359‡ | 1.6895‡ |
| Hallucination | GPT-4 | -9.0939‡ | 0.4145‡ |
| Success | Gemini 1.5 Pro | -8.7863‡ | 0.7397‡ |
| Hallucination | Gemini 1.5 Pro | -2.8503‡ | 0.2463‡ |

Table 6: Market cap analysis results on success and hallucination based on the empirical regression for GPT-4 and Gemini 1.5 Pro. *, †, and ‡ indicate significance at the 10%, 5%, and 1% levels, respectively. The study is conducted on RPD$_{200perYear}$ sample.

The results for Gemini and GPT-4 on RPD$_{200perYear}$ sample in Figure 10 and Table 6 are in accordance with the results of the main paper. Also the results for Llama-3-70B-Chat in Figure 11 are similar to the results for GPT-3.5 Turbo reported in Figure 5.

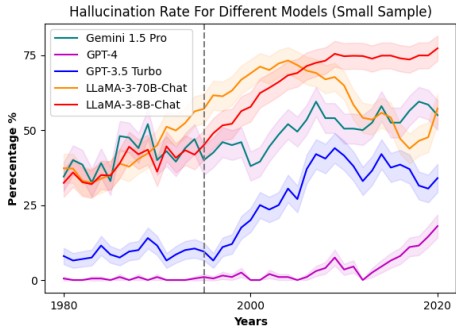

Figure 10: Hallucination of Gemini ("gemini-1.5-pro"), ChatGPT ("gpt-3.5-turbo-0613"), GPT-4 ("gpt-4-0613"), Llama-3-8B ("Llama-3-8B-Chat") and Llama-3-70B ("Llama-3-70B-Chat") over time. The dotted line is drawn at the year 1995. The performance is measured for all four models on $RPD_{200perYear}$ sample.

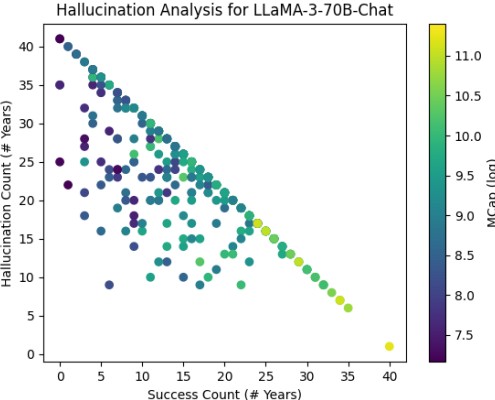

Figure 11: Each dot represents a company with a count on the x-axis indicating the number of years for which Llama-3-70B-Chat gave the correct answer and the y-axis indicating the number of years for which Llama-3-70B-Chat hallucinated in the answer. We take an average of market cap to assign a color of the dot.

## I    ROBUSTNESS CHECK: SAME COMPANIES OVER TIME

Although our experiments provide substantial evidence showing LLMs' proficiency in answering financial questions for more recent periods and larger market cap companies, it is imperative to consider the potential confounding factors. The bankruptcy and establishment of companies throughout the years could introduce variability, thus potentially affecting the outcomes. Figure 12 indicates LLMs' temporal performance on 460 companies that have existed consistently over the 41 years in our sample, aligning consistently with our previous comprehensive companies analysis. Eliminating the bias from bankruptcy and IPOs, we can assert that LLMs exhibit enhanced capability in recent periods. The reason for enhanced performance compared to the full sample can be attributed to survival bias(Elton et al., 1996; Rohleder et al., 2011).

In terms of market capitalization, we ran the same regression analysis on those companies. The result, displayed in Table 7, is similar to the previous market cap analysis. It reaffirms the claim that the larger the company's market cap is, the more accurate LLMs' answers to its financial questions are.

## J    WHY "REVENUE"?

Below we provide further reasoning on why we picked "revenue" to form our question prompts.

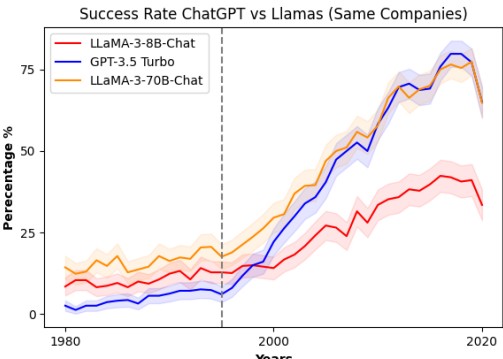

Figure 12: Performance of GPT-3.5 Turbo ("gpt-3.5-turbo-0613") and Llamas ("Llama-3-8B-Chat", and "Llama-3-70B-Chat") on $RPD_{460}$. The dotted line is drawn at the year 1995. The shadow area around the line is the standard deviation of model performance.

| Model | Constant ($\alpha$) | Beta ($\beta$) |
|---|---|---|
| GPT-3.5 Turbo | -12.6799‡ | 1.0247‡ |
| Llama-3-8B | -9.0914‡ | 0.7488‡ |
| Llama-3-70B | -8.2554‡ | 0.7245‡ |

Table 7: Market cap analysis results on $RPD_{460}$ over time based on the empirical regression. *, †, and ‡ indicate significance at the 10%, 5%, and 1% levels, respectively. The results are based on the full sample with year fixed effect.

- *Goal of our study*: As the goal of our study is to understand the temporal and cross-sectional biases, we want to ask questions that can vary across time as well as firms. Given our goal, it is not ideal to ask questions like "Can you explain what derivative securities mean?". The answer to this question is the same across time and firms.

- *Why not some financial ratio?*: Revenue is a top-line number that is directly available in SEC filings. If we form a question on a financial ratio like *return on assets*, even though we can validate the answer, it will involve two skills of model 1. ability to recall knowledge and 2. ability to calculate the ratio (do the arithmetic). As here we are focused on understanding the knowledge gap, a simple question on revenue is an appropriate choice.

## K    NEWS PREDICTION ANALYSIS

To understand how the cross-sectional bias can relate to the model's inability to assess the impact of news headlines on the stock price, we construct a prompt based on the news headline dataset created by Dong et al. (2024) and ask GPT-3.5 Turbo ("gpt-3.5-turbo-0613") to provide stock recommendation on whether to "BUY", "SELL" or "DNK" (do not have enough knowledge of the company). We filtered out the news headline dataset according to the knowledge cut-off time of GPT-3.5 Turbo[13]. For each company in our list in the year 2020, we chose a random (with random_state=1729) headline from our filtered news headline dataset.

We use the following prompt: *"Forget all your previous instructions. Pretend you are a financial expert with stock recommendation experience. Based on the following news headline give either BUY, SELL, or DNK (do not have enough knowledge of the company) recommendation in the first line and give a short reason in the second line. Headline: {headline}"*

We then evaluate whether the model's recommendation was correct or incorrect if the output label is not DNK. We consider the recommendation to be correct if the output is "BUY" ("SELL"), and there is a positive (negative) return over the next trading day. Otherwise, the recommendation is incorrect.

---

[13]We only keep news released on or after October 1st, 2021.

We run the same empirical regression used earlier in the paper. In this case, the Y variable is assigned a value based on whether the recommendation is correct, incorrect, or DNK.

| Model | Constant ($\alpha$) | Beta ($\beta$) |
|---|---|---|
| No Prediction (DNK) | 0.1175‡ | -0.0094† |
| Correct Prediction | 0.7472‡ | 0.0209‡ |
| Incorrect Prediction | 0.8097‡ | 0.0141* |

Table 8: Market cap analysis results on a stock recommendation based on the empirical regression for GPT-3.5 Turbo. *, †, and ‡ indicate significance at the 10%, 5%, and 1% levels, respectively. The study is conducted on a list of companies we have in the year 2020.

The result in Table 8 indicates that the model is more likely to make a decision (no matter whether it is correct or incorrect) for larger market-cap companies while more likely to provide the label "DNK" for small market-cap companies. The result serves as a tiny experiment to show how the cross-sectional bias can impact biases in investment recommendations. We note that this is just a study on a subset of news headlines but it can be extended and thoroughly evaluated on a large set of news headlines in a separate future study.

## L    SPANISH SOCCER LEAGUES STATISTICS STUDY

In order to demonstrate the applicability of our LLM bias evaluation framework to non-finance domains, we examined the application of our framework to assess LLMs' abilities to answer questions about soccer statistics. We compiled a dataset of 1,676 samples containing season statistics for a given season and team in both the La Liga and Segunda División Spanish soccer leagues. Our dataset spans 41 seasons from 1980 to 2020 and contains statistics about 167 unique teams, though not every team is present every year, as teams are relegated down to or promoted from lower leagues based on their performance.

We then assess the ability of Llama3-70B (*temperature*=0, *max_tokens*=100) to answer the question: "*In the {year} {La Liga/Segunda División} season, how many points did {team} finish with?*". We assess the quality of responses across both the temporal and cross-sectional dimensions, looking at differences in model performance in different years, as well as differences in model performance for clubs with greater prominence (measured by finishing position). Model performance is measured using a success rate metric, which for this study is defined as the percentage of model responses which give a point value within 5% of the true point value.

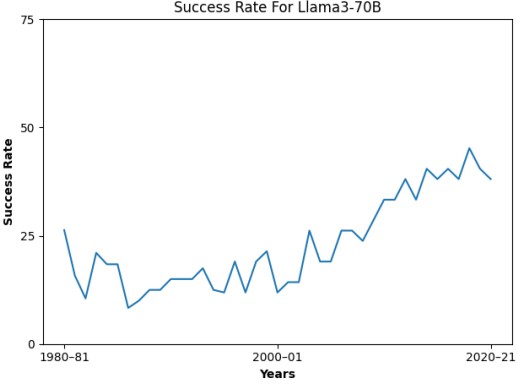

Figure 13: Success Rate of Point Answering Over Time.

Looking at the performance of the model, temporally (see Figure 13), we can see that the model performs better in more recent years, suggesting a bias towards more recent years.

Furthermore, we assess how the prominence of a soccer club affects the model's performance by running a logistic regression on $Pos_{i,t}$, the finishing position of each soccer club. Our logistic regression is defined in Equation 4.

$$logit(P(Y_{i,t} = y)) = \alpha + \beta * Pos_{i,t} + \delta_t * D_t + \gamma * \mathbb{1}_{\{league=Segunda\}} + \epsilon_{i,t} \qquad (4)$$

Here $Y_{i,t}$ is the outcome variable where $y = 2$ indicating model success, $\delta_t$ is a year-fixed effect, $\alpha$ is a constant term, $\gamma$ is a league-fixed effect, and $\epsilon_{i,t}$ is an error term. We find $\gamma$ to be $-3.1317$, reflecting the fact that being in Segunda División decreases the probability of getting the correct answer compared to La Liga.

We find $\alpha = -1.2941$ and $\beta = -0.0542$ [14], suggesting that clubs with a greater finishing position (lower-prominent clubs) see a lower success rate in our experiment. This highlights a knowledge bias stratified along team prominence, where the LLM is more likely to have knowledge of statistics regarding more prominent and successful teams.

## M  ETHICS STATEMENT

Our work adheres to ethical considerations, although we acknowledge certain biases and limitations in our study.

- *Geographic Bias*: Our study focuses solely on publicly listed companies in the United States of America; Our findings may not be fully representative of global firms and markets.

- *Data Ethics*: The data used in our study, which is derived from publicly available sources, does not raise ethical concerns. All raw data is obtained for public companies that are obligated to disclose information under the guidance of the SEC and are subject to public scrutiny.

- *Language Model Ethics*: The language models employed (with proper citation) in our research are publicly available and fall under license categories that permit their use for our intended purposes. While most models (Llamas) employed are publicly available, it is important to note that prompt answers of ChatGPT will be made public under OpenAI's terms of use. The terms of use of OpenAI do not allow the use of prompt outputs for building competing models. Given the nature of our data, we believe this condition does not diminish the use of our work. We acknowledge the environmental impact LLMs have but we believe that the impact from just inference is limited.

- *Dataset Ethics*: We will not make any raw data used for the project public but we will make all the revenue prompts (RPD) and their answers public on our GitHub repository.

---

[14]Significance at the 1% level

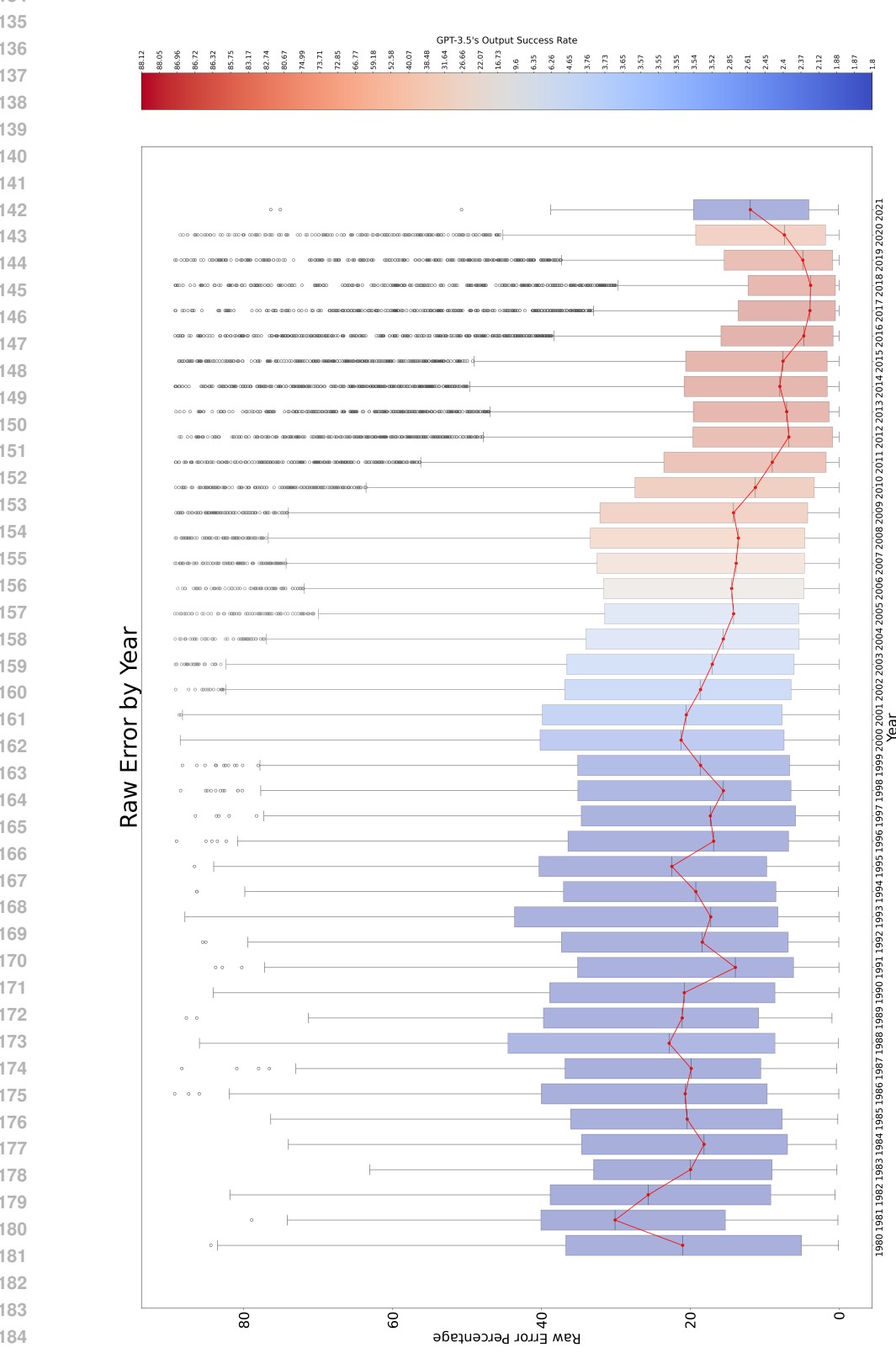

Figure 14: Raw answer error of GPT-3.5 Turbo ("gpt-3.5-turbo-0613") over time.

