# OpenReview forum: "Are Large Language Models Truly Democratizing Financial Knowledge? Identifying Knowledge Gaps"
_ICLR.cc/2025/Conference — Submitted to ICLR 2025_

### Official Review · Reviewer_QVuK · 2024-10-28

**Soundness:** 3
**Presentation:** 3
**Contribution:** 3
**Rating:** 6
**Confidence:** 3

**Summary:**

This work proposed a systematic method to analyze the knowledge biases of LLMs in the financial field and did a comprehensive evaluation through a relatively large dataset. Key findings indicate that LLMs show higher accuracy for more recent information and larger companies. The paper also highlights a tendency for LLMs to hallucinate more frequently for larger companies, even though these models tend to perform better with them.

**Strengths:**

1. The paper addresses an interesting topic on the bias of LLMs in financial domain.
2. The paper is clear. The analysis is based on relatively large datasets and the conclusions make sense.
3. The proposed framework can be adopted beyond financial field to evaluate LLM biases.

**Weaknesses:**

1. The related work only mentioned existing research regarding societal and political biases and hallucination in LLMs, it would be great to add some background on LLMs in financial domain.
2. As mentioned in the limitation section, using “revenue” as a proxy for questions might constrain the diversity of questions and knowledge.

**Questions:**

1. The results could be more convincing if the findings are consistent on other mainstream LLMs like Claude and Gemini.
2. Can we include some details on how significance is encoded in the regression method?

---

> ### Author Response · Authors · 2024-11-20
>
> Your review of our work is very constructive and helpful. We appreciate your insights into our work and your review, noting our generalizable framework for evaluating LLM biases. We address the weaknesses mentioned and your questions below.
>
> **Regarding more background on LLMs in the financial domain**: In our introduction, we mention several papers that highlight the efficacy of LLM usage across various financial use-cases. Based on your comments, we have further expanded our Section 6 “Related Work” to include further discussion of LLM usage and LLM performance in the financial domain.
>
> **Regarding Revenue Question**: We agree with the reviewer. Further, we would like to draw your attention to Appendix K, where we show the impact of the bias for trading with sophisticated prompt.
>
> **Question about Gemini**: We appreciate your feedback; we have incorporated Gemini results in Appendix Figure 8, Figure 10, and Table 6. The results are consistent with our findings.
>
> **Question regarding statistical significance**: We appreciate the careful reading of our paper. *, $\dagger$, and $\ddagger$ indicate p-value at the 10\%, 5\%, and 1\% levels, respectively for regression coefficients.

---

> > ### Comment · Reviewer_QVuK · 2024-11-29
> > **Response**
> >
> > Thank you for your response. I'm glad my comments will help you improve your paper.

---

### Official Review · Reviewer_mfyg · 2024-11-03

**Soundness:** 1
**Presentation:** 2
**Contribution:** 1
**Rating:** 1
**Confidence:** 5

**Summary:**

The paper finds that LLMs tend to retrieve corporate financial information when answering questions about recent periods and larger firms. However, it is more likely that LLMs hallucinate when they answer questions about larger firms and data from recent years. I applaud the authors for setting up a nice dataset for financial data retrieval. However, the paper has serious empirical and institutional issues, as stated below.

**Strengths:**

One potentially interesting aspect of this study is that LLMs tend to hallucinate more for larger firms. This, at first glance, seems counterintuitive because the authors find that LLMs’ accuracy is higher for larger firms. However, considering that retail investors are more likely to ask questions about larger firms, and if they get “wrong” answers, it is potentially a more serious problem than not being able to get any answers. Perhaps, one potential way to improve the implications of this study is to develop tools to minimize hallucinations in answering financial questions (see my comment below).

**Weaknesses:**

This paper lacks institutional details, and the authors do not seem to understand how language models are used in practice. Large brokerage houses and institutional investors never use an LLM to retrieve financial knowledge. In retrieval models, a specific context is explicitly given, and the models are fine-tuned to answer questions “within” this context window. They train their own models to improve performance in answering financial questions only after providing the model with a specific set of information. What the authors test in this paper is not how language models are used in practice or academia for financial research. Nobody in practice asks questions about firms’ fundamentals to an LLM. Instead, LLM’s potential lies in making economic predictions or interpretations given the financial fundamentals of a firm. Anyone can quickly look up the basic fundamentals of firms nowadays, and it is less likely that investors will use an LLM to retrieve financial information. To answer the question of whether LLMs are democratizing financial knowledge, the authors should look at economic interpretations of given financial information (i.e., whether LLMs can answer economic questions better than traditional platforms). I am not convinced at all that the authors answer the question of financial information democratization by simply asking GPT, “What is the revenue of firm A in year X?”.

Considering that we don’t know exactly which datasets LLMs are trained on, many studies have already explored LLMs’ performance based on recent knowledge. It is already well-known that LLMs tend to have better knowledge of recent events. The reason that LLMs answer questions about larger firms more accurately is potentially because they are covered more frequently by media articles, which are publicly available. As more electronic information has been produced in recent years, it is also trivial that LLMs answer questions about recent fiscal years more accurately. It is not surprising that LLMs do better for larger firms, and this finding does not make enough contribution to clear the bar at ICLR.

This paper is based on a very simple research design and I do not have much to add in terms of execution. However, I am unsure why the authors use only a simple one-shot prompt. Knowledge retrieval can benefit greatly from (i) allowing the model to search external knowledge and (ii) adding an additional step of validation. At least, the authors may want to try several techniques that are known to reduce hallucination (multi-agentic approach, few-shot prompts, etc.). Also, simply by adding “if you do not have a concrete source of information, give me N/A instead of writing up your answer” will help.

Your prompt does not consider the difference between fiscal year and calendar year. Most financial reporting is based on the fiscal year and writing a prompt like “What was the revenue of XX in 2002?” is confusing. Furthermore, the authors need to be more specific in that they require “annual” revenues.

**Questions:**

See "Weaknesses".

---

> ### Author Response · Authors · 2024-11-20
>
> Thank you for your feedback on our work. We appreciate your insight into our work, but we respectfully disagree with some of the comments. We respond to your comments below.
>
> **Response to misunderstanding of Use Case of LLMs in the financial industry**: The reviewer seems to misunderstand the motivation for the paper and how language models are used by retail investors in practice. We used “democratizing” in our title as our focus is not large investors.  Of course, analysts at Jane Street are not going to query LLMs for revenue information. We don’t believe nor expect large brokerages and institutional investors to use LLMs for retrieval of revenue related information that they can get from their paid data sources. Our study is designed keeping in mind 10s of millions of vulnerable retail investors. As we explained in our original submission (line #162-171), our experiments are not representative of LLM usage by industry experts; however, our study is motivated by how retail investors or other financial non-professionals may use LLMs. As also highlighted in our introduction, we are motivated by demonstrated retail investors’ reliance on LLMs, as we believe this population to continue to be vulnerable to the biases described in our paper. We concern ourselves with this population specifically due to recent claims that LLMs will “democratize financial knowledge” (Yue et al. 2023) and as such we wanted to highlight biases that may be harmful to unsophisticated investors participating in this “democratization.” These financial non-professionals are unlikely to use LLMs optimally, yet their investing activities are still consequential. We further highlight our concerns regarding non-professional investors in Section 5.
>
>
>
> Furthermore, our choice of prompt question is justified in Section 2.2. We highlight that while such knowledge-retrieval questions are not likely used in institutional financial settings, novice investors may employ such approaches and be susceptible to the biases presented in our paper. In addition, we chose our prompt to find a representative example of the type of knowledge question novice investors may ask LLMs; we did not aim to create a prompt that would capture all LLM use cases for investing.
>
> We also like to bring your attention to the Appendix J “WHY "REVENUE"?” which explains our rationale for why we use revenue as question.
>
> We would also like to bring your attention to Appendix K which shows the impact of the biases for trading and Appendix L which shows how the framework is also generalized to Spanish soccer league data.
>
> **Response to Temporal Exploration of LLM Performance**: Previous studies have examined the performance of LLMs on information produced after the cut-off date, yet to our knowledge there is no previous work on evaluating LLM biases across a large historical range. While we agree that larger media coverage and mentions in training data is likely responsible for both temporal and cross-sectional knowledge gap, we are the first, to our knowledge, to empirically confirm this hypothesis. Our introduction clearly talks about this.
>
> Also, contrary to the reviewer's point LLMs don't do better for larger firms as their hallucination rate is also higher for larger firms.
>
> **Response regarding Sophisticated Prompting**: Again, the reviewer seems to misunderstand the motivation for the paper and how language models are used by retail investors in practice. As highlighted before, we don’t expect retail investors and unsophisticated investors to use multi-agentic approach, and few shot prompts or other sophisticated investors to reduce hallucinations. As mentioned in Section 2.2, we focus our bias evaluation on replicating LLM usage associated with retail investors and/or non-financial professionals. This population is unlikely to use the LLM prompting techniques suggested, like few-shot prompting and multi-agent approaches, and as such, we omit such techniques from our bias evaluation. We also believe that “very simple research design” is a strength not a weakness of our work.
>
> **Response regarding Fiscal Year and Calendar Year**: We design our prompt to be representative of the type of prompt non-professional investors may use when using LLMs. As such, we omit nuances such as the differentiation between fiscal and calendar year in our prompt as we do not believe such details would be included in the prompt used by an retail investor. As we do not specify a quarter but specify a year, it is annual revenue.  Also, we believe that most retail investors are unaware of the difference between the calendar and fiscal year.
>
>
>
> We would like to also bring your attention to Appendix K which shows the impact of the biases for trading, and Appendix L which shows how the framework is also generalized to Spanish soccer league data.

---

> ### Comment · Reviewer_mfyg · 2024-11-20
> **Response**
>
> I am not satisfied with the authors' response. Please see my answers to your response below, point by point.
>
> 1. The authors claim retail investors primarily use GPT for financial metric retrievals, such as revenues and net income. This claim is unsubstantiated and contradicts existing surveys, which consistently show retail investors use GPT for investment advice rather than metrics easily accessible via search engines. See:
>
> https://www.investopedia.com/investors-increasingly-relying-on-chatgpt-to-manage-investments-7559052
>
> https://www.etoro.com/news-and-analysis/press-releases/one-in-ten-retail-investors-using-chat-gpt-style-ai-to-help-pick-and-manage-investments/
>
> I do not misunderstand the motivation of the paper. What I am not satisfied with is the authors’ execution. Retail investors, unlike the authors’ claim, do not generally use GPT to ask for revenues, net income, etc. It is the opposite in reality, and they ask for investment advice. If the authors want to claim that the primary purpose of GPT for retail investors is knowledge retrieval, provide me with any authoritative source or survey evidence. What I know from other studies and from large-scale surveys is different from what the authors claim.
>
> If the authors want to examine whether the technology is democratizing financial knowledge (and investment), they should assess the models’ ability to provide financial advice. This has already been tested in a large sample from many other papers.
> Sophisticated investors will use higher-quality prompts along with their professional judgment (i.e., human + AI), while less sophisticated investors are more likely to be susceptible to GPT’s recommendations. This is akin to what the authors do in Appendix K. However, Lopez-Lira and Tang (2023) have already done this analysis.
>
> 2. I am also not very convinced that end-users do not know anything about prompting. Even under restrictive assumptions of zero-shot prompting, GPT models integrated with web search significantly enhance performance. For example, using the authors' Figure 1 prompt, GPT-4 retrieved NextEra Energy's 2006 revenue ($15.71 billion) via web search. Did the authors' API include search functionality? If not, the evaluation omits a crucial capability of real-world implementations.
>
> 3. Appendix J's focus on revenues rather than net income undermines the paper's relevance to financial performance analysis. Net income is the standard performance metric in finance. Testing irrelevant contexts (e.g., "Spanish soccer league") detracts from the paper's focus. A tighter domain-specific approach is needed to strengthen the analysis.

---

> ### Comment · Reviewer_mfyg · 2024-11-20
> **Response 2**
>
> 4. While the authors claim novelty in demonstrating GPT's temporal and cross-sectional knowledge gaps in finance, similar findings are well-documented in broader LLM literature (e.g., Bender et al. 2021). Many papers have already shown that less-represented events in the training data can cause hallucinations and inaccurate answers. Surprisingly, the authors do not cite any of these papers. Of course, judging a paper’s contribution is a subjective issue. But extending this to finance, while potentially interesting, lacks sufficient novelty for a general-interest venue like ICLR.
>
> Additionally, the conclusion that GPT hallucinations occur more for larger firms but are also more likely to provide answers is unclear. What is the takeaway? If the authors suggest LLMs are unsuitable for financial knowledge retrieval, the contribution seems limited to documentation. You might want to take your study to develop a knowledge-retrieval tool for retail investors. But again, using LLMs for financial metric retrieval does not seem like an ideal use case.
>
> 5. The authors argue again that retail investors cannot distinguish between fiscal and calendar years. However, then it becomes even more confusing how the authors evaluated their model performance. The model will surely know the difference between fiscal and calendar years. If you ask for the 2006 revenue, are you referring to the revenue released in 2006 or the revenue of fiscal year 2006? How does the model interpret this question? Either interpretation can be correct. How did you evaluate the “correctness” of the model’s answer?
>
> Again, the authors argue that I misunderstood their motivation. I perfectly understand their motivation. However, the authors' execution and institutional understanding are far from ideal and cannot clear the bar at ICLR.
> Overall, the authors’ response did not resolve any of the issues raised in my previous report. I also reviewed other reviewers’ comments and agreed with the reviewer zyuC. The paper lacks novelty and does not provide convincing empirical evidence.
>
>
> References
>
> Bender, Emily M., et al. "On the dangers of stochastic parrots: Can language models be too big?🦜." Proceedings of the 2021 ACM conference on fairness, accountability, and transparency. 2021.
>
> Lopez-Lira, Alejandro, and Yuehua Tang. "Can chatgpt forecast stock price movements? return predictability and large language models." arXiv preprint arXiv:2304.07619 (2023).
>
> Kim, Alex, Maximilian Muhn, and Valeri Nikolaev. "Financial statement analysis with large language models." arXiv preprint arXiv:2407.17866 (2024).
>
> Fieberg, Christian, Lars Hornuf, and David J. Streich. Using gpt-4 for financial advice. No. 10529. CESifo Working Paper, 2023.
>
> Niszczota, Paweł, and Sami Abbas. "GPT has become financially literate: Insights from financial literacy tests of GPT and a preliminary test of how people use it as a source of advice." Finance Research Letters 58 (2023): 104333.
>
> Niszczota, Paweł, and Sami Abbas. "GPT as a Financial Advisor." Available at SSRN 4384861 (2023).

---

> > ### Author Response · Authors · 2024-12-04
> > **Response to Reviewer Follow-Up**
> >
> > **Response to 1.**: We are happy that after our rebuttal, the reviewer realized that the study is focused on retail investors and not large financial institutions as they seem to have originally perceived. We would like to further clarify that we do not mean to suggest that retail investors use LLMs purely for knowledge retrieval. However, they rely on the knowledge of LLMs implicitly. When retail investors ask LLMs for investment advice, the LLMs’ investment advice is based on their own knowledge of the company including financials. As such, the biases that we study in our paper become relevant; the LLM may give worse financial advice for firms of which it has less knowledge.
> >
> > We read the work of Lopez-Lira and Tang (2023), and at least in the current version, they don’t demonstrate the biases of LLMs for large vs small firms.
> >
> >
> > **Response to 2.**: Again, we are happy to see that after our rebuttal, the reviewer is not suggesting that retail investors would be using a “multi-agentic approach” for prompting as they seem to have originally perceived. At the time of the original submission of this paper, GPT-4 did not feature web search functionality. Furthermore, we feel that web search functionality is not a part of the LLM itself, and that it is not appropriate to include it in evaluations. Also, companies like OpenAI can add/remove features on top of LLMs over time. In our view, GPT-4 with web search is a product, not LLM.
> >
> > **Response to 3.**: We believe that the demonstration of our bias evaluation framework on non-finance domain data demonstrates its applicability to other domains; we aim to develop a framework that will allow for analysis of LLM biases across a variety of strata in a variety of domains as discussed in the “Broader Impacts and Future Work” paragraph in Section 5. We believe that the demonstration of similar biases for different domains is a strength and not a weakness of our work. The number of papers at top conferences that conduct similar analysis is testament to this.
> >
> > **Response to 4.**: In our view, Bender et al. 2021 paper opens a line of research on LLM biases and is not the end of the literature.  In addition, our framework for bias evaluation can help identify those specific under-represented stratas in various domains. We believe it is crucial for LLM users in various domains, including finance, to be aware of their domain-specific biases. While it may be trivial to say that less-represented events in training data lead to worse model performance, it is not always trivial to identify what those less-represented events, or stratas, are. Our framework makes it easier to identify these events, which is especially crucial as LLMs see applications in an ever-broader set of domains. Again, we would like to highlight that the finding that “LLMs are more likely to hallucinate for more prominent entity” is novel.
> >
> > **Response to 5.**: We would like to clarify that for most companies the calendar and fiscal year are the same. In our view it is more likely for the retail investors to ask for revenue or some other financial metric for a given year rather than including the term “fiscal year”.

---

### Official Review · Reviewer_zyuC · 2024-11-04

**Soundness:** 2
**Presentation:** 3
**Contribution:** 2
**Rating:** 3
**Confidence:** 4

**Summary:**

This paper examines the breadth and biases in large language models' (LLMs) knowledge concerning financial data on U.S. publicly traded companies. Using a dataset of over 190,000 questions, the study assesses how well LLMs represent historical financial information and explores the influence of company characteristics (like size, market cap, and institutional attention) on model performance and hallucination rates. Findings indicate that while LLMs are more accurate with data on larger firms and recent information, they also tend to hallucinate more frequently for such cases, revealing inherent biases that challenge the notion of democratized financial knowledge in LLMs.

**Strengths:**

Addresses a key issue: The paper tackles an important topic by assessing the biases and limitations of large language models in representing financial knowledge, specifically for U.S. publicly traded companies. This focus on knowledge gaps in LLMs and its impact on democratizing financial information is timely.

**Weaknesses:**

- Lack of Novelty: The paper lacks innovation in methodology. It primarily repurposes existing techniques to evaluate LLMs without introducing new methods or frameworks. While the dataset itself is new, the analytical approach does not present methodological advancements. Additionally, the evaluation focuses on a limited selection of three general-domain LLMs without a comprehensive comparison to specialized financial LLMs like FinMA, FinGPT, or other representative LLMs (e.g., GPT-4, GPT-4o, Gemma, Mistral). The narrow scope limits the generalizability and depth of the conclusions drawn.

- Insufficient Case Studies and Error Analysis: The study relies heavily on metric-based performance evaluations without deeper case studies or qualitative error analysis. Incorporating detailed case studies would enhance understanding of specific error patterns and biases in the models’ financial responses, providing more substantiation for the findings.

- Potential Copyright Concerns: The paper does not address potential copyright issues related to the financial dataset used for evaluation. More clarity is needed on the data source's legal and ethical aspects, particularly for research purposes.

- Introduction and Related Work Needs Reorganization: The introductory sections lack depth and miss key relevant background on many existing research in financial LLMs and LLM evaluation frameworks. A more comprehensive review of related work, particularly on financial LLM development and evaluation methodologies such as [1][2], would strengthen the contextual understanding and help to better position this work within the current landscape.

[1]  Is chatgpt a financial expert? evaluating language models on financial natural language processing. arXiv preprint arXiv:2310.12664 (2023).

[2] The finben: An holistic financial benchmark for large language models. arXiv preprint arXiv:2402.12659 (2024).

[3] Fineval: A chinese financial domain knowledge evaluation benchmark for large language models. arXiv preprint arXiv:2308.09975 (2023).

**Questions:**

See the weakness

**Details Of Ethics Concerns:**

The paper does not address potential copyright issues related to the financial dataset used for evaluation. More clarity is needed on the data source's legal and ethical aspects, particularly for research purposes.

---

> ### Author Response · Authors · 2024-11-20
>
> Thank you for your review of our paper, we appreciate your insight and suggestions, and we address the weaknesses you have brought up below.
>
> **Response to Lack of Novelty**: In our paper, we aim not to evaluate the performance of LLMs in the financial domain, but to identify biases in performance. The mentioned papers do not perform an analysis of how the models perform across various stratifications of the financial domain, which was our goal in this paper. We believe that our temporal and stratified approach to evaluating performance biases is novel and has not yet been done in the financial domain or any other domain, as mentioned in lines #100-102. We also show the broad applicability of our bias evaluation framework on soccer (Spanish club) data in Appendix L.
>
> **Response to Limited LLMs Evaluated**: We evaluate FinMA (which ranks last in performance among the models tested) in Appendix G of our paper and evaluate GPT-4 on a smaller sample (due to cost constraints) in Appendix F. We find that the general-domain models outperform FinMA significantly. We have added Gemini results in Appendix Figure 8, Figure 10, and Table 6. The results are consistent with our findings.
>
> **Response to Insufficient Case Studies and Error Analysis**: We present an example of the biases we study in Figure 1.
>
> **Response to Potential Copyright Concerns**: The reviewer probably didn’t notice our discussion of the sources of our data in Section 2.1 (for revenue data), Section 4.1 (for cross-sectional variables data) and further in Appendix B. Our RevenuePromptDataset is novel because it contains both prompts and LLM Responses for our revenue questions, which is data that was indeed produced because of our work; it does not simply include revenue data sourced from our mentioned data sources. We use LLM-generated data from proprietary models (e.g. GPT) in accordance with applicable “terms of use”, as highlighted in Appendix M. We also specifically note the public availability of our data sources in Table 1.
>
> **Response to Related Work**: The goal of our study was not to evaluate the overall performance of LLMs, but rather their biases along temporal and finance-specific stratifications. As such, our evaluation targets differ from those prioritized in previous financial LLM evaluation benchmarks. However, we have further expanded our Section 6 “Related Work” to include further discussion of LLM usage and LLM performance in the financial domain.

---

### Author Response · Authors · 2024-11-20
**Rebuttal by Authors**

We sincerely thank all the reviewers for their time. We are encouraged that the reviewers appreciate the following aspects of our work:

- Reviewers found the paper well-written, clear, and supported by a large and comprehensive dataset, which strengthens the validity of the analysis and conclusions. (QVuK)



- Our study tackles a timely and important issue by examining the breadth, biases, and limitations of large language models (LLMs) in representing financial knowledge for U.S. publicly traded companies. This focus on knowledge gaps and its implications for democratizing financial information has been highlighted as a significant contribution. (zyuC, QVuK)



- We proposed a systematic evaluation framework, leveraging a dataset of over 190,000 questions, to analyze how company characteristics (e.g., size, market cap, and institutional attention) influence LLM performance and hallucination rates. The framework has been noted as robust, with potential applications beyond the financial domain. (zyuC, QVuK)



- The paper presents key findings that LLMs perform better on larger firms and recent information but also exhibit higher hallucination rates under these conditions. This nuanced insight into the trade-offs in LLM performance has been recognized as both counterintuitive and impactful, especially given its relevance to retail investors. (zyuC, QVuK)



We have carefully considered all feedback and believe the revisions based on Reviewer QVuK’s comments have enhanced the quality of the paper and its contributions. Below, we provide individual responses to each reviewer's comments. As Reviewer QVuK provided particularly insightful and constructive feedback, we have prioritized addressing their suggestions in the updated PDF.

---

### Meta-Review · Area_Chair_8wqG · 2024-12-21

**Metareview:**

This paper examines bias in LLMs in the financial domain. The proposed evaluation framework leverages a dataset of over 190k questions to analyze how company characteristics influence LLM performance and hallucination rates. With a primary focus on knowledge gaps and its implications for democratising financial information, the paper presents interesting findings: LLMs perform better on larger firms and recent information but also exhibit higher hallucination rates under these conditions.

The paper addresses bias in the financial domain: an important and timely topic with real world relevance.

The paper is well structured and clearly written.

Wider applicability of the proposed bias evaluation framework beyond the financial domain.

**Additional Comments On Reviewer Discussion:**

Lack of Novelty: the authors address this satisfactorily by pointing out the temporal and stratified approach to evaluating performance bias has not yet been done in the financial domain. I do agree with the authors that the paper’s contribution does present novel insights.

Limited selection of LLMs evaluated: the additional evaluation performed (reportedly consistent with other findings in the paper) and incorporated into the manuscript address this concern.

Concerns raised around copyright, details on significance in the regression method, and related work have all been addressed satisfactorily.

More fundamentally, disagreements surrounding the motivation of the work were not fully justified or satisfactorily addressed. And unfortunately, the validity and significance of this paper, to some extent, rests on the clarity and consistency of these conceptual issues.

---

### Decision · Program_Chairs · 2025-01-22

Reject